# Enhancing Core Public Service Vocabulary to Enable Public Service Personalization

**Alexandros Gerontas** [1,*] **, Dimitris Zeginis** [2] **, Rafail Promikyridis** [1] **, Marin Androš** [3] **, Efthimios Tambouris** [1] **, Vibor Cipan** [3] **and Konstantinos Tarabanis** [2,4]

1   Department of Applied Informatics, School of Information Sciences, University of Macedonia, 156 Egnatia Street, 54636 Thessaloniki, Greece; r.promikyridis@uom.edu.gr (R.P.); tambouris@uom.edu.gr (E.T.)
2   Information Technologies Institute, Centre for Research and Technology Hellas (CERTH), 6th km Harilaou—Thermi, 57001 Thermi, Greece; zeginis@iti.gr (D.Z.); kat@iti.gr (K.T.)
3   RIDE Technologies d.o.o., Vrhovcev Vijenac 40, 10000 Zagreb, Croatia; marin@pointjupiter.com (M.A.); vibor@pointjupiter.com (V.C.)
4   Department of Business Administration, School of Business Administration, University of Macedonia, 156 Egnatia Street, 54636 Thessaloniki, Greece
*   Correspondence: agerontas@uom.edu.gr

**Abstract:** The provision of public services (PS) is at the heart of public authority operations as it directly affects citizens' lives and the prosperity of society. Part of PS provision is publishing PS descriptions in an online catalogue to inform citizens and promote transparency. The European Commission has developed Core Public Service Vocabulary Application Profile (CPSV-AP), as a standard European PS data model to facilitate PS catalogue creation and semantic interoperability. However, CPSV-AP is not sufficient to model complex PS with different versions based on rules and citizens' circumstances (e.g., getting a passport for a child or for an emergency). As a result, citizens cannot obtain personalized information on PS. The aim of this paper is to enhance CPSV-AP in order to support the modeling of complex PS. We illustrate the use of the proposed model in a real-life case study. Specifically, we use the proposed model to develop a knowledge graph and a chatbot that provides personalized information to citizens of the city of Bjelovar (Croatia) regarding the life-event "having a baby". We believe our research is of interest to researchers on PS data models and public authorities interested in providing personalized PS information to their citizens using chatbots.

**Keywords:** public service; personalization; knowledge graph; chatbot; CPSV-AP





## 1. Introduction

The provision of public services (PS) is an important aim of public authorities worldwide. PS provision implements governmental policies to fulfill citizens' needs. An important part of PS provision is publishing structured information about PS (also known as PS descriptions) in PS catalogues. This empowers citizens, contributes to open government, and promotes transparency and trust between public administrations and citizens. Furthermore, active participation of citizens in PS improvement is facilitated, thus promoting PS co-creation [1].

PS descriptions are based on an underlying PS model [2]. In this article, the term PS model is used to denote "a data model that has been developed for describing and/or developing public services" [2]. A PS model includes concepts and may also include relationships between them. Examples of concepts often used in PS models include title, description, legal framework, cost, etc. Universally accepted PS models are considered as main enablers for promoting semantic interoperability and quality in PS provision.

Embracing the need for a standard PS model, the European Commission (EC) has developed the Core Public Service Vocabulary (CPSV) [3]. The first CPSV version was released in 2013. Subsequently, a linked data application profile of CPSV (namely CPSV-AP)

has been developed. The more recent version of CPSV-AP, i.e., version 2.2.1, was published in 2019 [4]. CPSV-AP belongs to a set of core vocabularies [5] that have been elaborated by the EC in the framework of ISA/ISA$^2$ programmes. This set of core vocabularies comprises, apart from CPSV, the Core Criterion and Core Evidence Vocabulary (CCCEV), the Core Person Vocabulary (CPV), the Core Business Vocabulary (CBV), etc. The adoption of CPSV-AP and other EC ISA/ISA$^2$ core vocabularies by the Member States is expected to facilitate semantic interoperability and the implementation of cross-border public services.

CPSV-AP constitutes a solid model for describing simple PS. However, in reality, many PSs are complex, having different versions that correspond to different citizens' profiles or other circumstances [6]. A complex PS may have many different versions according to different citizens' profiles (e.g., married or not married) and needs (e.g., urgent issuance of a passport). For different versions of a complex PS, citizens, for example, may have to submit as evidence different documents or may have to pay a different amount. Modeling PS versions has been identified in the academic literature as an important challenge [6]. This modeling issue hampers the provision of personalised information about a PS and increases the administrative burden. Clearly, obtaining personalised information becomes particularly important in the case of complex PSs. In summary, although CPSV-AP is a promising standard, recent research has shown that further enhancement is needed in order to accommodate PS personalization [2,7,8].

Additionally, previous research has shown that the use of knowledge graphs for storing CPSV-AP based PS descriptions can facilitate obtaining personalised information about PSs [9]. Knowledge graphs can store CPSV-AP based PS descriptions and, in addition, employ rules to correlate different concepts of PS versions (e.g., if the citizen is over 18 years old then the cost is €80 else it is €60). However, acquiring information directly from a knowledge graph is not an easy task for citizens.

Lately, chatbots have been increasingly used in PS provision for implementing a rich communication channel with citizens [10]. Recent research has shown that using chatbot technology on top of CPSV-AP based linked data can facilitate the structured provision of information about PSs in a user-friendly and intuitive way [7,11]. The use of chatbots on top of CPSV-AP based data that have been organised and stored as knowledge graphs seems very promising for the provision of personalised information about PSs in a user friendly way [12]. However, there is limited research work on this topic.

In this context, the aim of this paper is two-fold: (a) to enhance CPSV-AP in order to support PS personalisation by reusing classes of other EC ISA/ISA$^2$ core vocabularies (namely Core Criterion and Core Evidence Vocabulary, Core Person Vocabulary and Core Business Vocabulary), as well as classes that have been identified in the literature, and (b) to demonstrate the use of enhanced CPSV-AP through a pilot implementation using a chatbot on top of enhanced CPSV-AP based data that have been organised and stored as a knowledge graph.

The rest of the paper is organised as follows. Section 2 presents background and related work about EC ISA/ISA$^2$ core vocabularies, knowledge graphs, and chatbots. Section 3 presents the methodology used in this paper, while Section 4 presents the proposed enhanced CPSV-AP model. Section 5 applies the model in the case of the life-event "having a baby" in the city of Bjelovar (Croatia) and presents a pilot implementation based on knowledge graphs and chatbots. Finally, Section 6 discusses the results and Section 7 provides conclusions and directions for future work.

## 2. Background and Related Work

In this section, some background information about EC core vocabularies, chatbots, and knowledge graphs is provided. Additionally, related work on the use of chatbots and knowledge graphs in PS provision is outlined.

## 2.1. Public Service Versions and Life Events

A PS can be defined as "a mandatory or discretionary set of activities performed, or able to be performed, by or on behalf of a public organisation, publicly funded, and arising from public policy" [4]. A PS may correspond to the provision of a benefit to a citizen or business, e.g., to the parents of a newborn baby, or to the fulfillment of an obligation, e.g., the issuance of an identity card.

Some PSs are simple, e.g., the issuing of a birth certificate. For simple PSs, the required input (e.g., documents), cost, output, etc. are the same for all citizens. In other words, all concepts of a PS model used for modeling a simple PS have the same values for all citizens. On the contrary, some PSs are complex. Here, the required input, cost, etc. depend on the citizens' profiles and needs [6]. A citizen that is eligible for a complex PS will use a particular version of that complex PS according to their particular profile and needs. For example, two different citizens that need the same PS might be asked to submit a different set of documents or to pay a different amount. The specific values of all concepts of a PS model constitute a complex PS version. Usually, the appropriate version of a PS is determined based on a set of questions that a citizen is asked either by a public servant or an online dialogue system. As a result, the number of versions of a complex PS might be very high depending on the number of questions and possibly answers per question. In summary, citizens always acquire personalised information that match their exact needs. However, this is not always possible as underlying PS models cannot support complex PSs.

When providing information to citizens, public authorities often use the metaphor of a life event (LE). For citizens, a LE describes "situations of human beings where public services may be required" [13]. Citizens in different stages of their lives have different needs, e.g., to study, to start a professional activity, to get married, to have a baby, to travel, etc. For public authorities, a LE refers to "the government services needed at specific stages in life" [14]. LEs act as a bridge between citizens and public administration. A LE is implemented as a group of PSs that are executed in a specific order to fulfill citizens' needs [6]. For example, "having a baby" is a LE that may include a group of PSs, e.g., "birth registration", "maternity allowance", etc. As a LE is a set of PSs, personalisation of a LE is related with the personalisation of the included PSs. More specifically, the specific versions of the PSs of a LE that match the profile of a citizen constitute a specific version of a LE.

## 2.2. EC Core Vocabularies

This section presents the EC ISA/ISA$^2$ core vocabularies, namely the Core Public Service Vocabulary Application Profile (CPSV-AP) (Section 2.2.1), Core Criterion and Core Evidence Vocabulary (CCCEV) (Section 2.2.2), Core Person Vocabulary (CPV), and Core Business Vocabulary (CBV) (Section 2.2.3), that are exploited for deriving the enhanced CPSV-AP.

### 2.2.1. Core Public Service Vocabulary Application Profile

The UML class diagram of CPSV-AP 2.2.1 is depicted in Figure 1. CPSV-AP is a data model for describing PSs and the associated life and business events [4]. The classes and properties of CPSV-AP are categorized as mandatory or optional. The mandatory classes (i.e., Public Service class and Public Organization class) have blue colour, while optional classes have orange colour. According to CPSV-AP, a PS requires some evidence (e.g., documents or data) in order to be executed. The required evidence is based on relevant legal resources. After the execution of a PS, an output (e.g., a certificate) is produced. A PS is provided through a communication channel (e.g., a mobile application) and has a cost. A CPSV-AP concept is the Event class. Subclasses of the Event class are the Business Event and the Life Event. An example of a Business Event is the establishment of a new business while an example of a Life Event is having a baby.

### 2.2.2. Core Criterion and Core Evidence Vocabulary

The EC introduced the Core Criterion and Core Evidence Vocabulary (CCCEV) v1.0.0 in 2016 [15] and v2.0.0 in 2022 [16]. CCCEV v2.0.0 contains two basic and complementary core concepts, namely the Core Criterion and the Core Evidence. Core Criterion models the requirements for the provision of a PS while Core Evidence models the evidence, including information, documents, or other, for fulfilling those requirements. CCCEV is designed to support the exchange of information between organisations or persons, which are specific cases of the concept Agent. The UML class diagram of CCCEV v2.0.0 is depicted in Figure 2.

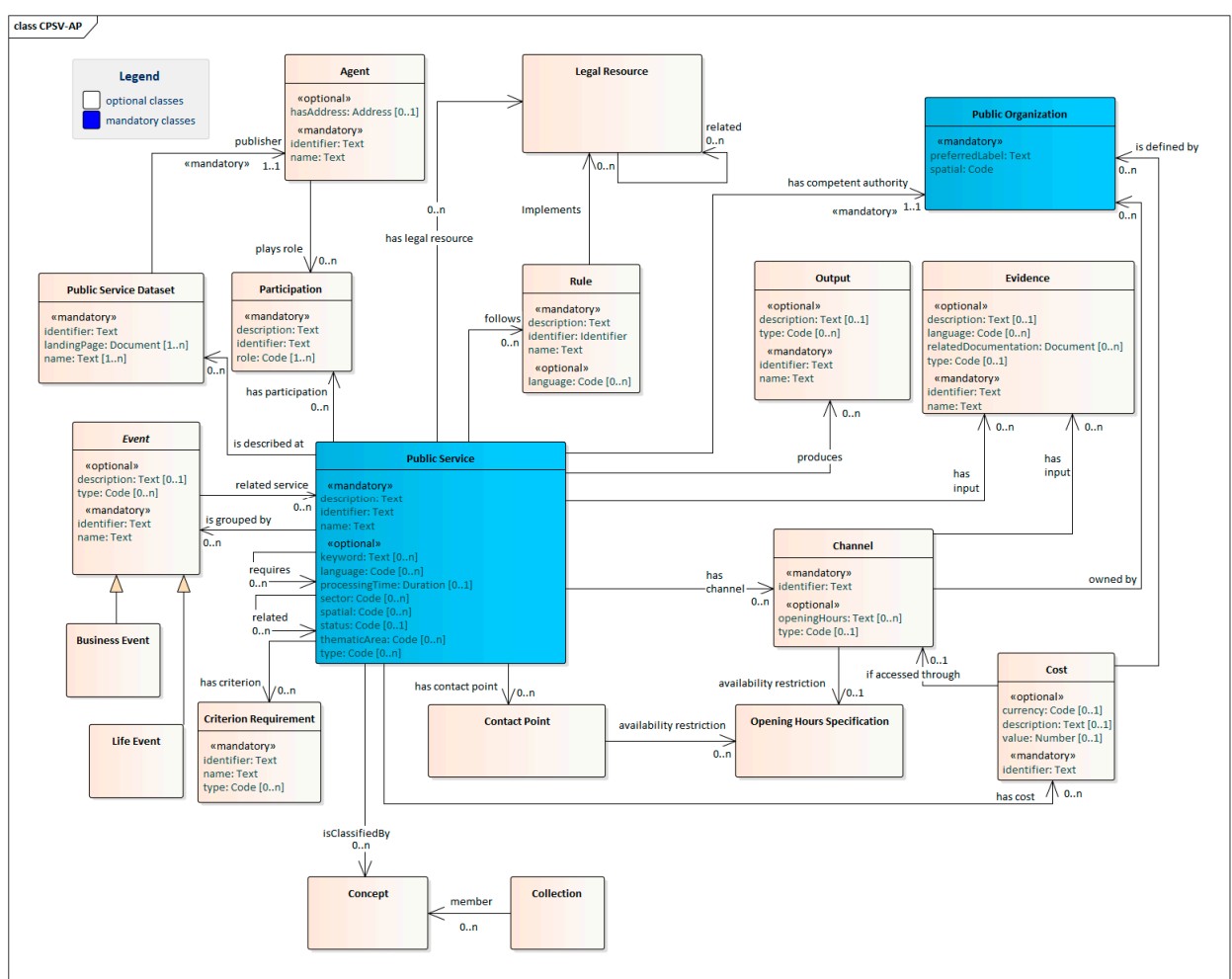

**Figure 1.** UML diagram of CPSV-AP 2.2.1 (adopted from [4]).

### 2.2.3. Core Person and Core Business Vocabularies

On 1 April 2021, the EC published a working draft of Core Person Vocabulary (CPV) [17] and Core Business Vocabulary (CBV) [18]. CPV provides a minimum set of classes and properties for describing a natural person, i.e., the individual as opposed to any role that they may play in society or the relationships that they may have with other persons or organisations. All classes and properties of CPV constitute the broader concept of identity [17]. The CBV provides a minimum set of classes and properties for providing information about legal entities, i.e., "trading bodies that are formally registered with the relevant authority and that appear in business registers". Sole traders, virtual organisations, and other types of Agents that are able to do business, are excluded [19].

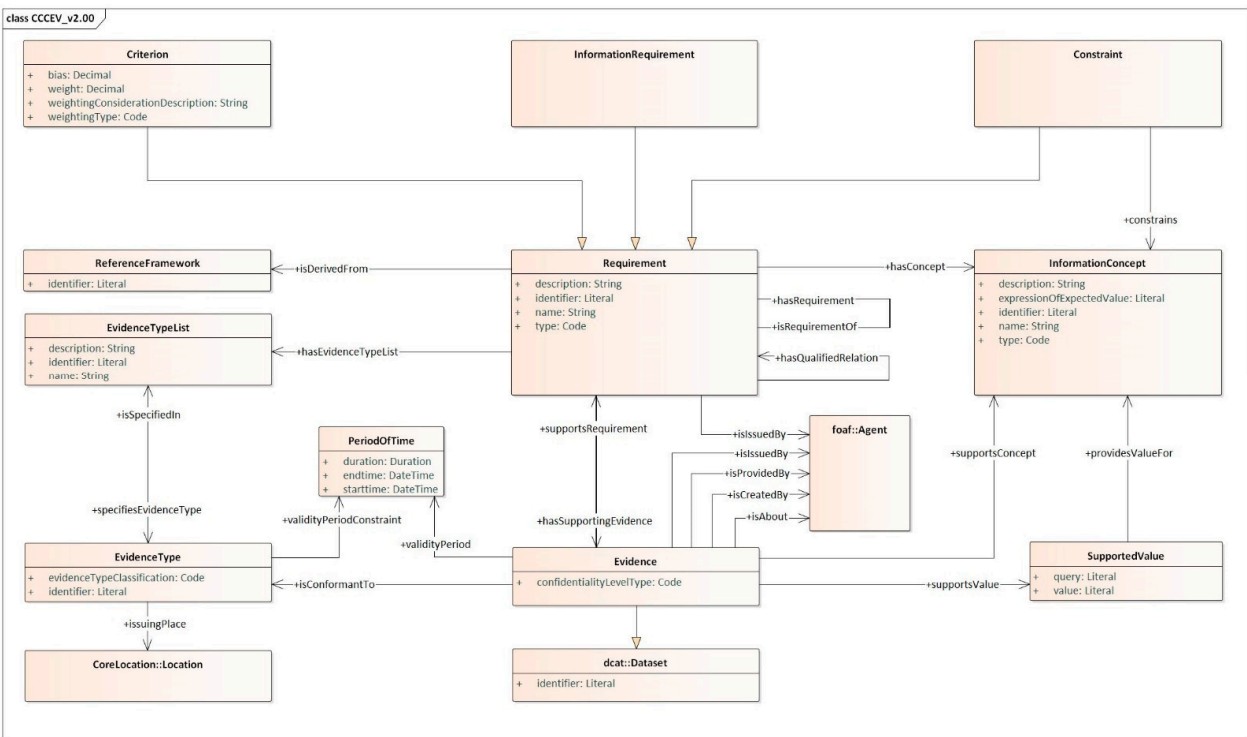

**Figure 2.** UML diagram of Core Criterion and Core Evidence Vocabulary v2.0.0 (adopted from [16]).

### 2.3. Chatbots and Knowledge Graphs

In this subsection, we review related work on knowledge graphs and chatbots technologies that can contribute towards personalization in the PS provision domain.

Google, with the creation of its own knowledge graph, triggered global interest in the term knowledge graph [20]. Although there is no commonly accepted definition by the community, according to [21] "A knowledge graph acquires and integrates information into an ontology and applies a reasoner to derive new knowledge". In other words, knowledge graphs allow the representation of knowledge using the structure of a graph. A knowledge graph consists of four main parts: (a) Nodes are the entities and represent the distinct objects of the modeled domain; (b) Edges are the relations and describe how two or more entities are connected; (c) Attributes describe different properties of the entities; (d) Rules allow reasoning of the data and the derivation of new information. For example, deductive reasoning can be used based on conditions and new knowledge can be generated as a result.

Currently, knowledge graphs are mainly used in the private sector. However, there is also some research on exploiting knowledge graphs in the public sector. Recent research includes the development of a knowledge graph for the PS 'Get a passport' in Greece [9]. That knowledge graph was based on CPSV-AP model and its goal was to provide citizens with information about required documents regarding all possible PS versions according to a citizen's profile. Another knowledge graph was developed based on CPSV-AP enriched model [2]. The aim of that knowledge graph was to investigate the potential benefits of its use to manage PS information based on domain-specific rules [22]. These rules can facilitate the provision of personalised information on PS versions. In both cases the ontology was CPSV-AP model. As CPSV-AP does not care for machine readable rules, the rules of knowledge graphs are employed. Both knowledge graphs were developed using Grakn.ai (https://vaticle.com/ (accessed on 20 April 2022)) software (now renamed Vaticle), which is an open-source software platform for knowledge graph development.

Chatbots are artificial intelligence applications that are widely used in the private sector to provide personalized information through natural language dialogue with the user, e.g., Amazon's Alexa, Apple's Siri, etc. In a common chatbot design, the input of the user is analysed using NLP methods. Then, the results of the analysis are matched

with predefined patterns (training data by developers), making the intentions of the user understandable to the chatbot. In the last step, based on the intentions of the user, the chatbot answers in natural language with predefined phrases [23]. The communication between the chatbot and the database is usually performed using APIs. Chatbots use has been increasing in many domains, such as education [24], health [25], business [26], etc. Lately, there has also been an increase in their use in the PS provision domain aiming to improve communication with citizens and to provide information online through a user-friendly interface [27].

For example, the city of Vienna introduced the WienBot chatbot that aims to help with citizen's questions asked on the city page about the online available services. The chatbot provides answers to frequently asked questions and by using it, citizens can have direct access to the information they need without having to search various pages [28]. Another example is the UNA chatbot in Latvia. UNA's purpose is to answer the frequently asked questions regarding the procedures of enterprise registration. Citizens can be informed about the registration of their business and its progress by using the chatbot [29]. Moreover, in Ukraine, there are several chatbots that have been implemented with success [30].

Furthermore, a few articles use CPSV-AP to model PS descriptions and publish them as linked data. The produced linked data repositories, containing structured PS descriptions, are exploited for providing information about PSs through chatbots [7,11,31]. An extension of CPSV-AP that facilitates chatbot implementations has also been proposed [32].

Finally, in [12], chatbots are applied on top of knowledge graphs. The aim of the chatbot was to provide information about the issuance of a passport. The chatbot was developed using the software Rasa (https://rasa.com/ (accessed on 20 April 2022)), which is an open-source tool for building custom AI chatbots using Python and natural language understanding (NLU). The connection of the chatbot with external APIs is achieved thanks to the Rasa action server, which allows the execution of custom actions. Dialogue engine predicts which custom actions to execute. For the communication between the chatbot and the knowledge graph, the Grakn client interface is deployed through a Rasa custom action. The Grakn client is responsible for database operations.

## 3. Methodology

The methodology followed is inspired by the principles of design science research methodology (DSR) [33]. DSR begins by identifying current problems that need to be solved or potential opportunities for improvements in an actual application environment consisting of people, organizational systems and technical systems [34].

Our methodology aspires to enhance CPSV-AP in order to be able to support the provision of personalised information on PSs. It comprises the following steps:

(1) Present the latest version of CPSV-AP (As-Is situation) and the relevant Core Vocabularies, i.e., CCCEV, CPV, and CBV (Section 2), and evaluate CPSV-AP regarding its ability to support PS personalization (Section 2).

(2) Present related work on CPSV-AP, knowledge graphs, and chatbot technologies (Section 2).

(3) Enhance CPSV-AP by reusing classes from other EC ISA/ISA$^2$ core vocabularies or classes that have been derived from related work (Section 4).

(4) Develop a pilot implementation for demonstrating the use of enhanced CPSV-AP (Section 5).

(5) Discuss the effectiveness of enhanced CPSV-AP for supporting PS personalization based on the pilot implementation (Section 6).

## 4. The Enhanced CPSV-AP Model

This section presents the enhanced CPSV-AP model. The aim of the enhancement is: (i) to "merge" CPSV-AP and CCCEV in order to facilitate providing personalised information on PSs, as well as (ii) to include other related concepts (Feedback and Potential

Consumer) that have been identified in the literature [2]. The Potential Consumer concept is incorporated into the enhanced model by reusing classes from CPV and CBV.

The enhanced CPSV-AP model is depicted in the UML diagram of Figure 3. In blue color, the classes of the existing CPSV-AP are depicted (some classes are omitted for simplicity), while in green color the classes of CCCEV are depicted, in purple the Core Person and Core Business Vocabulary classes, and in orange the other related classes that have been identified in the literature. Table 1 presents the classes of the enhanced CPSV-AP, the vocabulary they belong to, their description, and the rationale for their inclusion in the CPSV-AP enhancement.

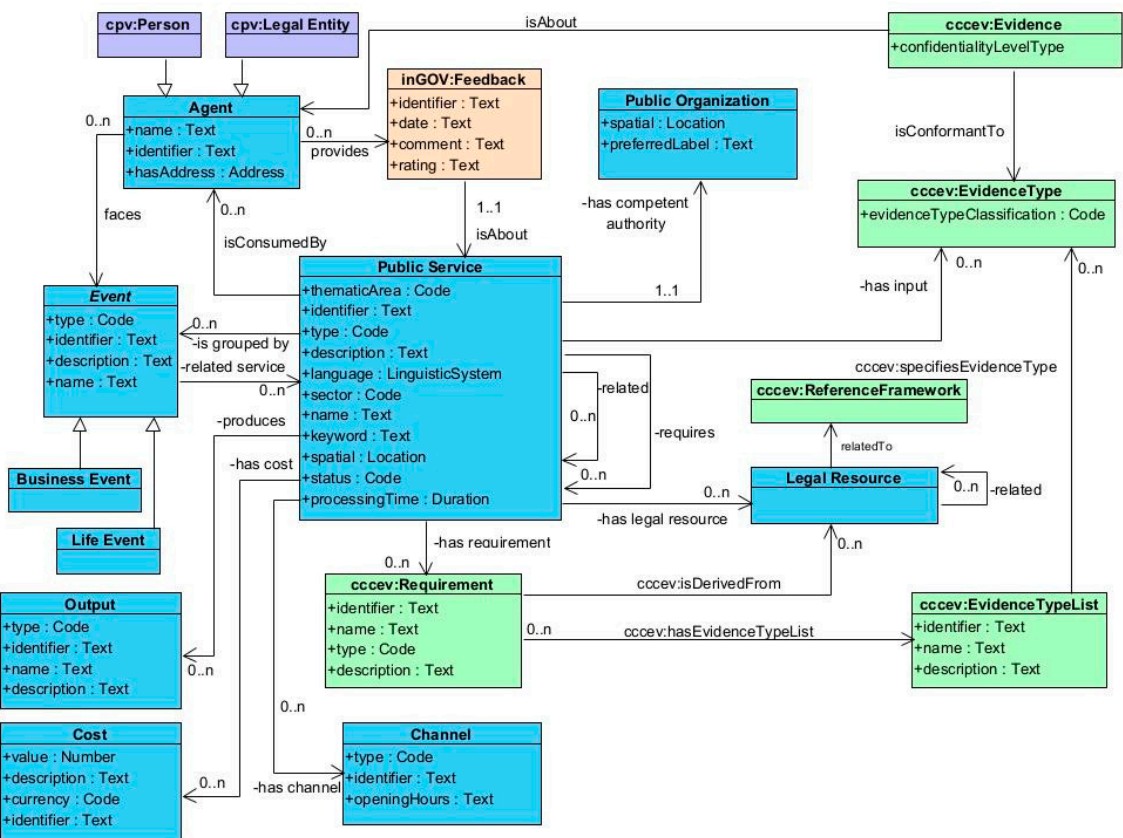

**Figure 3.** The main classes of the enhanced CPSV-AP 2.2.1 model.

In order to achieve PS personalization, we focus on CPSV-AP classes that may vary amongst PS versions. These include (i) Output class: different PS versions may produce different outputs, e.g., passport with validity of 5 years and passport with validity of 1 year; (ii) input, modeled through the class EvidenceType e.g., different inputs may be needed for EU and non-EU citizens; (iii) Channel class: the PS could be executed through different communication channels (e.g., online service, physical office); (iv) Cost class, e.g., the execution of the PS at the physical office may impose extra costs; and (v) Requirement class: different PS versions have different requirements, e.g., one version may be executed only by EU citizens and another only by non-EU.

The enhancements of CPSV-AP proposed in this paper are described in the following paragraphs. The current version of CPSV-AP includes a Criterion Requirement class and an Evidence class which however are not associated. Consequently, the inference of the necessary Evidence that addresses a Criterion Requirement of a PS is not straightforward. Toward this direction, in the proposed enhancement, the Criterion Requirement class of CPSV-AP is substituted by the Requirement class of CCCEV, while the Evidence class of CPSV-AP is substituted by the EvidenceType class of CCCEV. At CCCEV, there is a direct connection between the Requirement and the EvidenceType, enabling in this way their

association. Additionally, other classes of CCCEV are also reused (i.e., EvidenceTypeList, Evidence), providing a richer conceptualization.

**Table 1.** Enhanced CPSV-AP classes.

| Class | Vocabulary | Description | Extensions & Rationale |
|---|---|---|---|
| Public Service | CPSV-AP | Represents the Public Service itself, as it is described in a public service catalogue | Core mandatory class of CPSV-AP for describing PS and their versions |
| Public Organization | CPSV-AP | The responsible Agent for the delivery of a PS | Core mandatory class of CPSV-AP for describing the provider of a PS |
| Event | CPSV-AP | Represents an event (Business or Life Event) related to a set of PS | Groups related PS enabling their integration |
| Legal Resource | CPSV-AP | The legislation, policy or policies that lie behind the Rules that govern the service | CCCEV and CPSV-AP use different classes to represent the legislation that defines the rules that govern a PS. Their association (relatedTo) is an extension proposed in this paper. |
| Reference Framework | CCCEV | Legislation or official policy from which Requirements are derived. | |
| Output | CPSV-AP | Any resource, document, artifact produced by the PS | Output may vary between PS versions |
| Cost | CPSV-AP | Any costs related to the execution of a PS that the consuming Agent needs to pay | Costs may vary between PS versions |
| Channel | CPSV-AP | Represents the medium through which an Agent interacts with a PS e.g., online service, phone | Channels may vary between PS versions |
| Requirement | CCCEV | Condition or prerequisite that is to be proven by Evidence | The association (has requirement) between the PS class & the Requirements is an extension proposed in this paper. Requirements may vary between PS versions. |
| Evidence Type List | CCCEV | Group of Evidence Types (all of them should be present) for conforming to a Requirement | The same Requirement between different PS or PS versions should be addressed by the same Evidence Type List (s) enabling PS interoperability and integration |
| Evidence Type | CCCEV | Information about the characteristics of an Evidence | The association (has input) between the PS class and the Evidence Type is an extension proposed in this paper. |
| Evidence | CCCEV | Proof that a Requirement is met | The Evidence is specific for an Agent that consumed the service |
| Agent | CPSV-AP, CPV, CBV | Represents the potential consumer of the service and can be a Person or a Legal Entity | The association (is consumed by) of the PS class with the Agent is an extension proposed in this paper |
| Feedback | from literature | Enables users to provide feedback at the information and execution phases of PS | Fosters co-creation by actively involving PS consumers in PS provision. This is an extension proposed in this paper |

Another association between CPSV-AP and CCCEV classes that is proposed in this paper includes the linking of the Legal Resource (CPSV-AP) and Reference Framework (CCCEV) classes. They both refer to the legislation that defines the rules that govern PS provision. This linking enables interoperability between CPSV-AP and CCCEV.

Another enhancement includes the direct association of the Public Service class with the Agent class defining the consumer of the Public Service (this is a gap already identified in the literature [2]). The current version of the CPSV-AP could express this association indirectly by exploiting an intermediate class Participation. However, the role of the PS consumer is fundamental in PS provision, and thus it makes sense to add it explicitly in the model. The consumer of the PS can be either a person or a legal entity (e.g., business). Thus, two corresponding classes from CPV and CBV have been added as subclasses of the Agent class.

Finally, an Agent could potentially provide Feedback about a PS either at the information phase (i.e., when searching for available information for a PS) or the execution phase (i.e., during/after the execution of a PS). The provision of Feedback is currently not supported by CPSV-AP. Consequently, Feedback has been added and is related to the

Agent class and to the Public Service class. Feedback could be structured, e.g., 5-star rating, or unstructured, e.g., comments in free text.

## 5. Applying the Model: The Case of Newborn Public Service—City of Bjelovar

In this section, the enhanced CPSV-AP model is applied in the context of the Newborn PS that is offered by the city of Bjelovar in order to model PS information as a knowledge graph (Section 5.1). Additionally, a chatbot is developed (Section 5.2) on top of the enhanced CPSV-AP based knowledge graph for the provision of personalised PS information to citizens.

### 5.1. The Newborn Public Service

The city of Bjelovar (Croatia) offers the "Newborn PS" that handles the child's registration in the Registry Office. The "Newborn PS" is grouped under the "Having a baby" life-event that also includes the following PSs: (a) Financial aid for the newborn's equipment from the Croatian Health Insurance Fund, (b) Registration of a newborn child on the tax card, (c) City's financial aid for a newborn child, (d) County's financial aid for a newborn child. All of the latter mentioned PSs have as a prerequisite the execution of the "Newborn PS".

In the registration form of the "Newborn PS", the parents are asked: (i) a set of questions, such as the child's name, parents' citizenship, etc. and (ii) a set of questions that allow the inclusion of other PSs of the "Having a baby" life-event, e.g., "Do you want to apply for financial aid from the county?", "Do you want to register a child on a parent's tax card?", etc. The answers to these questions impose the use of different versions of the "Newborn PS" that vary regarding cost, output, input, channel and requirements.

There are 128 different variations based on the answers to the questionnaire. For example: (i) if one of the parents has no Croatian citizenship, then the required input documents are different; (ii) if parents don't own an e-personal ID card, they cannot utilize e-services and have to go to Bjelovar's registry office directly (different channel); (iii) the direct use of the service through the registry office imposes further charges for processing the request.

Additionally, the answer to some of the questions may lead to the inclusion of another PS (e.g., financial aid for the newborn's equipment from the Croatian Health Insurance Fund). In this case, the Croatian "Health Insurance Fund PS" is a dependent service, while the "Newborn PS" is an independent service as it does not require the execution of another service in order to be executed.

In this pilot, we employ the enhanced CPSV-AP to model information about the "Newborn PS" and all PSs that are included in "Having a baby" life event in the form of a knowledge graph. A part of the knowledge graph is presented in Figure 4 that models the main "Newborn PS" as follows:

- The "Newborn PS" is modeled as a "Public Service".
- The "City of Bjelovar", that is the competent authority for providing this PS, is modeled as a "Public Organization".
- The "Newborn PS" is formally described in a legal framework that comprises a set of laws. All these laws, e.g., "Newborn PS legislation", are modeled as "Legal Resources".
- The "Newborn PS" is grouped under the life event "Having a baby". Another service under this life event is the "Financial aid PS".
- The "Financial aid PS" requires the successful execution of the "Newborn PS".
- An eligibility criterion of the "Newborn PS"is that the applicant must be a parent of the newborn. This is modeled as a Requirement associated with this PS. This Requirement is also derived from the legal framework. We should note that more eligibility requirements are included in the requirements list of the "Newborn PS", e.g., both parents should give their consent. However, for simplicity, here, we model only one requirement.

- The above Requirement can be fulfilled by different types of evidence, e.g an identi-
fication (ID) card, a birth certificate, etc. The set of evidence types required to fulfill
a requirement constitute an EvidenceTypeList. In our simple example, the required
EvidenceTypeList contains just one "Evidence Type", namely the ID card, which can
be used as a proof that the applicant is the parent of the newborn and thus fulfills that
specific requirement.
- The potential PS consumer, which uses the "Newborn PS", is modeled as Person.
- The potential PS consumer provides a specific Evidence, e.g., Maria's ID card, in order
to fulfill the specific Requirement of this example. The Evidence provided should be
conformant to an "Evidence Type" that is specified in an EvidenceTypeList.
- The service consumer may provide Feedback for a PS either at the information phase,
i.e., before using the PS, e.g., mentioning some missing information of the PS de-
scription, or at the execution phase, i.e., during/after using the service, e.g., stating
her/his satisfaction.

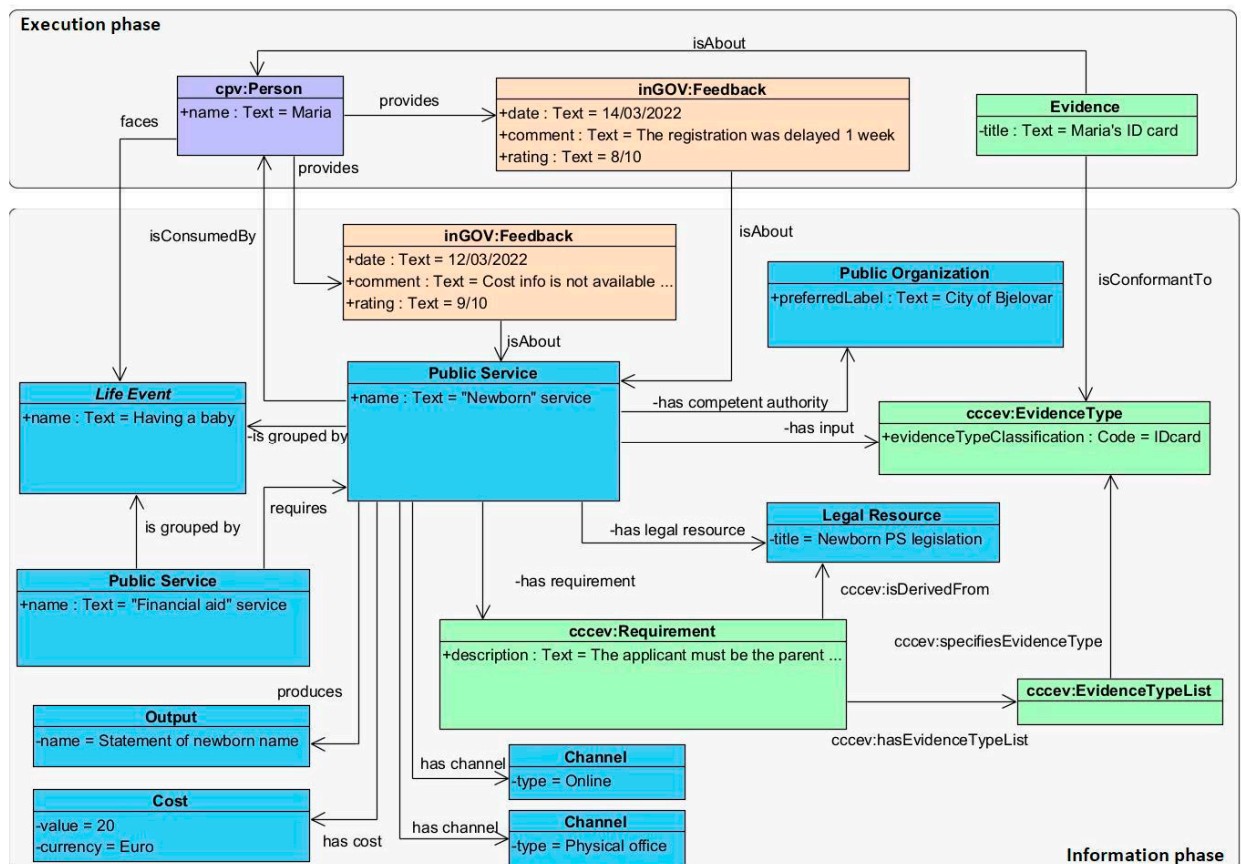

**Figure 4.** "Newborn PS" example using the enhanced CPSV-AP model.

### 5.2. Chatbot Implementation

This section describes the chatbot implemented on top of the developed knowledge
graph (KG). The chatbot aims at providing personalized information to the citizens of Bjelo-
var potentially serving more than 40,000 people. The personalized information includes:
(i) the version of the "Newborn PS" that the user is eligible for, including the cost, output,
input, channel, and requirements, and ii) other services that the user can benefit from.

#### 5.2.1. Overall Chatbot Architecture

The overall system architecture (Figure 5) consists of four layers:

- "Contact Channels layer" contains possible integrations of the chatbot solution to
third-party software (e.g., facebook messenger, whatsapp). This layer is responsible

for allowing a user to send messages to the chatbot and also display results from the chatbot to the user.

- "Presentation Layer" includes the chatbot engine that incorporates Natural Language Understanding (NLU) and Natural Language Processing (NLP) functionalities. The chatbot engine is responsible for processing and classifying inputs from the user messages. It is responsible for sending appropriate responses to the user. This layer is also responsible for formatting and presenting information received from the "Business Layer".
- "Business Layer" includes an API that links the KG database to the chatbot engine and exposes its data. The API is responsible for handling requests from the chatbot engine, i.e., to fetch data from the KG, format it, and forward it to the chatbot engine.
- "Database layer": is the KG database that is modeled based on the enhanced CPSV-AP and contains descriptions of PS (e.g., "Newborn PS"), life events (e.g., "Having a baby"), as well their relations.

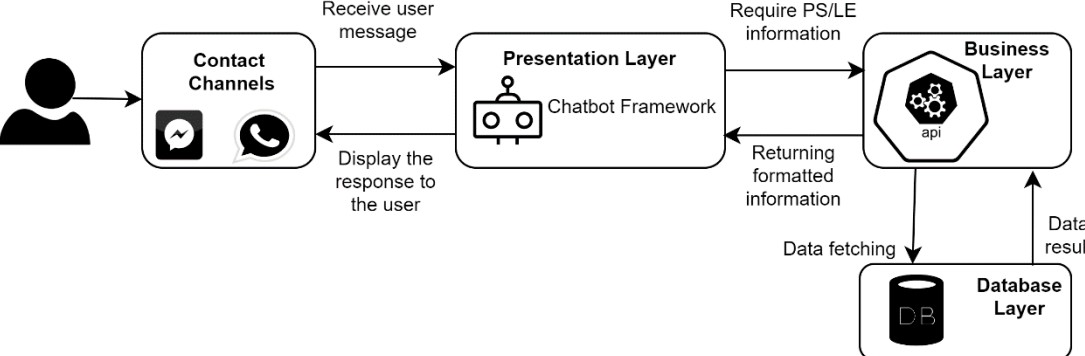

**Figure 5.** Overall chatbot architecture.

5.2.2. Implementation Decisions

In order to implement the architecture, open-source, free software solutions were used. The selection of software solutions is based on specific criteria as described in the following paragraphs. The criteria were derived from the existing needs of the City of Bjelovar that hosts the chatbot.

Regarding the "Presentation layer", the core technology used is the chatbot and needs to meet the following criteria: (i) support for Croatian language, (ii) support social network integration (e.g., Whatsapp, Facebook Messenger), (iii) provide the ability to interact with external REST APIs. Rasa open-source framework was chosen as a framework that fulfills most of the above requirements. Rasa supports NLP and NLU and can communicate with outside services (e.g., REST APIs) through the Rasa Actions Server. Rasa does not natively support the Croatian language. However, it offers the ability to plug-in third party NLU pipelines. Towards this direction, Stanza (https://stanfordnlp.github.io/stanza/ (accessed on 20 April 2022)) was used, which is a Python NLP package providing downloadable pre-trained NLP models for 66 languages including Croatian.

The "Business Layer" includes an API to facilitate the communication between the chatbot engine and the database. The API needs to support the implementation of security measures, be highly modifiable, and be modeled in the form of a REST API. The API was implemented in Node.js and exposes HTTP endpoints to allow data fetching from the DB.

Finally, the "Database layer" is realized by a Knowledge Graph DB that needs to meet the following requirements: (i) support the discoverability using life events, (ii) support the storage of data based on the enhanced CPSV-AP. The software that has been used is TypeDB (previously known as Grakn) which is a strongly typed database in the form of KG. TypeDB allows the storage of PS and life event descriptions based on the enhanced CPSV-AP as well as their interconnections using rules which are a part of TypeDB schema. The schema

of the Knowledge Graph DB also uses some chatbot specific classes as described in the next section.

### 5.2.3. Chatbot Specific Enhancements of CPSV-AP

Question-answer dialogues are the core of chatbot implementations. The existing CPSV-AP model does not support such question-answers dialogues to obtain the required information by users, thus we further enhanced CPSV-AP (Figure 6) to also address this need. This chatbot-specific enhancement includes the addition of the "Information" class which models the questions that a user is asked by the chatbot as well as the relevant answers. The answers to these questions affect: (i) costs, outputs, channels, and requirements that define a PS version that will finally be provided as a response by the chatbot back to the user and (ii) the inclusion of other related PS grouped under the same life event.

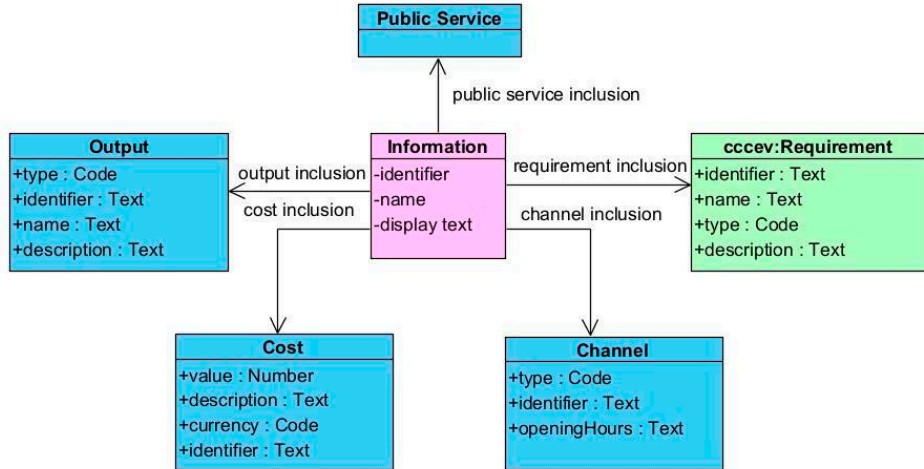

**Figure 6.** Chatbot specific enhancement of CPSV-AP.

It is important to note that the chatbot implementation assumes that every PS has a default set of costs, outputs, and requirements for its utilization. Based on the information items provided through the chatbot dialogue extra cost, output, requirement, and channel are appended to the default. This final set of cost, output, requirement, and channel correspond to a specific PS version. Finally, we should note that the different inputs (i.e., Evidence Type) that may be needed by different PS versions can be derived through their link with the Requirement class. In this case, the answer to a chatbot question will derive some "Information" that leads to the inclusion of a Requirement that is in turn associated with an "EvidenceType". That is why the "EvidenceType" is not explicitly associated with the "Information" class.

### 5.3. Usage Scenario

The scenario can be summarized as follows: "A citizen just had a newborn child and thus needs to register her/him and also benefit from related PSs included in the life-event "Having a baby"." The developed chatbot provides the following functionalities to address citizen's needs:

- Recognizes that the user wants information about the "Newborn PS". (Figure 7, Step 1)
- Retrieves all Information (i.e., questions that are stored based on the Chatbot specific enhancements of CPSV-AP presented in Section 5.2.3) related to this PS, in order to build the questionnaire to be used. The resulting data are structured in a single logical unit and formatted appropriately before being shown to the user (Figure 7, Step 2).
- Presents questions to the users and based on their answers gathers additional costs, outputs, channels, evidence types, and other related PSs that are related to the "Newborn PS" (e.g., the Financial aid for the newborn's equipment from the Croatian Health Insurance Fund PS). (Figure 7, Steps 3–8)

- Presents to the user personalised information about the version of the "Newborn PS" they are eligible for, as well as information about other relevant services. In the backend, this step retrieves the "default" cost/output/channel/evidence type for the PS and combines them with additional cost/output/channel/evidence type/PS that are imposed by the answers to the questions. (Figure 8)

- Enables the user to ask about all available services that the chatbot knows and get a list of them. Every PS on the list is a clickable item, making it easier for users to select one of them.

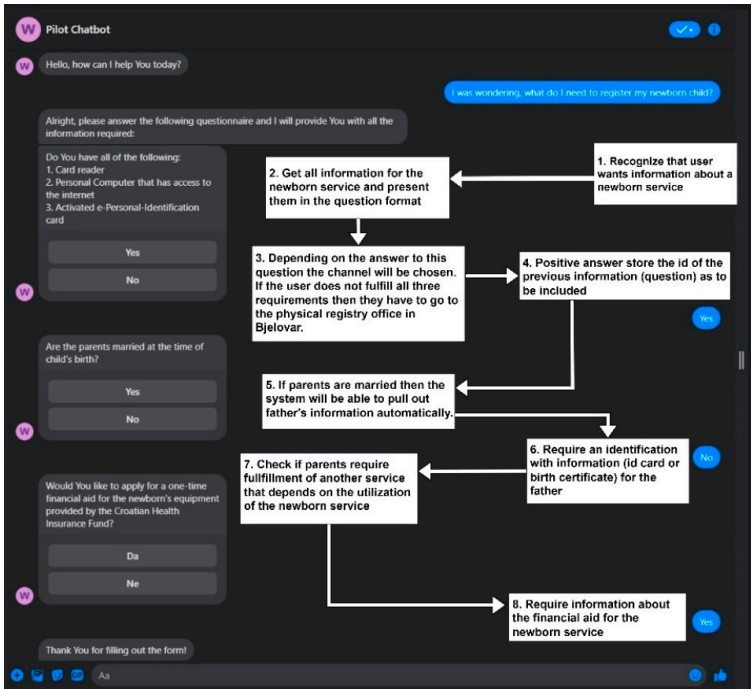

**Figure 7.** Chatbot dialogue excerpt for the "Newborn PS".

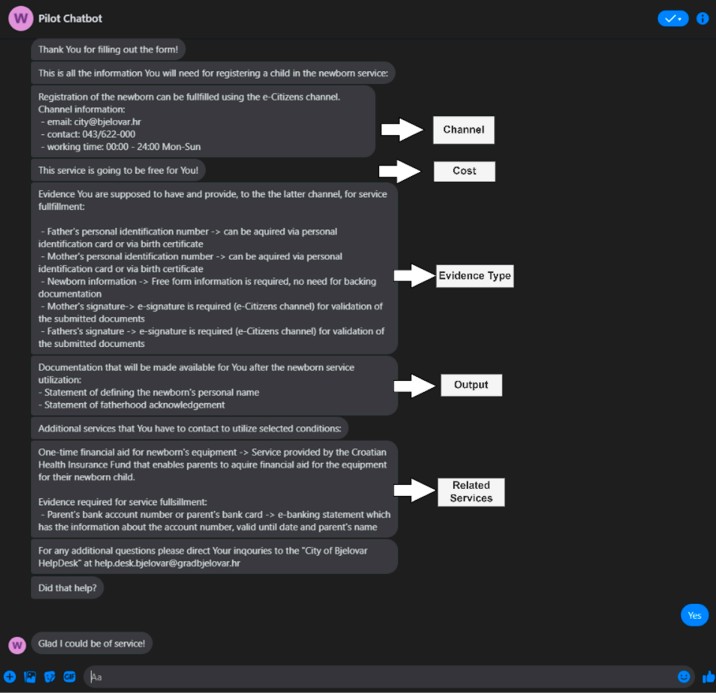

**Figure 8.** Personalized information provided by the Chatbot for the "Newborn PS".

## 6. Discussion

The proposed enhancement of CPSV-AP improves the ability of the model to provide personalised information. However, the proposed enhancement per se does not implement the business logic required in order to map citizens' profiles to specific PS versions. The mapping, and thus the personalisation, is realised by also exploiting knowledge graph and chabot technologies. The profile of the citizen is progressively created based on the information collected through a dialogue (questions/answers) of the citizen with the chatbot, which is then mapped to a specific PS version.

The proposed approach partially addresses personalisation also at the Life Event level. As a life event is a set of PSs, its personalisation is related to: (i) the identification of the subset of PS that belong to the life event and address the needs of a citizen and (ii) the mapping of the citizen's profile to the appropriate version of the identified PSs. However, the personalisation of life events is out of the scope of this paper and remains to be explored.

A potential improvement could be the use of knowledge graph rules to infer all relationships between "Information" class and "Cost", "Output", "Channel"and "Requirement" as well as the data that need to be gathered and appended to the query. This approach could add business logic to the knowledge graph level.

A possible holistic solution to the PS versioning issue might come up through the Rule class of CPSV-AP. The Rule class could potentially provide the mechanism for selecting the appropriate version of a PS. An obstacle toward this direction is that the CPSV-AP does not envisage instances of the Rule class as machine-readable business rules and the detailed modeling of the rules related to Public Services is out of scope of the CPSV-AP.

The proposed enhancement of CPSV-AP has been demonstrated in the case of "Newborn PS" of the city of Bjelovar. This service was selected as an example since it is relatively complex, comprising 128 different versions. Other PSs can be modeled in a similar way by instantiating the model and using all/subset of the classes/properties. In some cases, some of the classes may not be needed, e.g., some PS may have no cost so the "Cost" class will not be instantiated in such services. In order to implement the business logic required for the personalisation of other PSs, the "Information" class should also be instantiated and associated with the corresponding "Cost", "Output", "Channel", and "Requirement" that constitute a PS version.

Finally, we compare the model proposed in this paper with CPSV-AP and relevant CPSV-AP extensions proposed in the literature. Although CPSV-AP is a promising standard, it does not support personalisation in public service provision. Additionally, the extension of CPSV-AP proposed in [2] models the "Potential Consumer" of a PS introducing an ad hoc class. Finally, the extension proposed in [32] facilitates chatbot implementations by modeling the question–answer dialogue to obtain the information required by users. The extension of CPSV-AP proposed in this paper builds upon and improves the aforementioned extensions by: (i) adopting the concepts proposed in [2], but expressing them using standard models (i.e., CPV, CBV), and (ii) adopting and simplifying the chatbot related concepts proposed in [32] (i.e., in [32] the authors use two classes the "Question" and "Answer" while the model proposed in this paper uses only the "Information" class to model the chatbot-user interactions).

## 7. Conclusions

This article proposes an enhancement of CPSV-AP in order to enable the personalisation of PSs. The enhancement reuses classes from other EC ISA/ISA$^2$ Core Vocabularies, namely CCCEV, CPV, and CBV, and incorporates classes that have been identified in the literature and not included in CPSV-AP. Additional classes from CCCEV could be exploited if that will be deemed necessary in a future usage scenario.

The proposed enhancement of CPSV-AP has been applied in the case of Newborn Public Service in the city of Bjelovar, Croatia, also including a pilot implementation based on knowledge graphs and chatbots. The initial evaluation of the pilot implementation indicates that the use of knowledge graph and chatbot technologies is promising in realising

the personalisation of PS in real-life scenarios. However, their use in large scale scenarios remains to be explored. The current functionality of the chatbot supports a number of PSs grouped by the "Having a baby" life event. However, we aim to further extend this functionality to support more PSs and life events in order to cover a broad range of PSs offered by the city of Bjelovar.

Finally, we anticipate that the proposed enhancement of CPSV-AP will facilitate the creation of a knowledge graph of public services that will enable the development of advanced AI applications (including chatbots) for the provision of personalised and integrated services to citizens.

**Author Contributions:** Conceptualization, A.G., D.Z., R.P. and E.T.; formal analysis, A.G. and D.Z.; methodology, A.G., D.Z., E.T., and K.T.; software, M.A. and V.C.; validation, R.P., M.A. and V.C.; writing—original draft, A.G., D.Z., R.P. and M.A.; writing—review & editing, E.T. and K.T. All authors have read and agreed to the published version of the manuscript.

**Funding:** This work was funded by the European Commission, within the H2020 Programme, in the context of the project inGov under Grant Agreement Number 962563 (https://ingov-project.eu/ (accessed on 20 April 2022)).

**Data Availability Statement:** Not applicable.

**Conflicts of Interest:** The authors declare no conflict of interest.

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
