# Peer review of "Enhancing Core Public Service Vocabulary to Enable Public Service Personalization"

_information, doi:10.3390/info13050225_

Round 1

Reviewer 1 Report

The topic is very interesting and it corresponds to the new trends in public services -chatbots technologies, AI. 

The realisation of the chatbot for the The Newborn Public Service of the city of Bjelovar is described in depth and the results are presented in a scientific manner. This aproach could be used and implemented in other public services and another life events.

The paper is very well structured, readable and written in good English. I have founded some little mistakes in the text and I uploaded a file with highlighted words.

The diagrams are not readable. Is it possible to increase the font in the diagrams and graphs?

Reviewer 2 Report

This article aims to enhance the Core Public Service Vocabulary (CPSV-AP) in order to support the personalisation of Public Services by reusing classes of other EC core vocabularies, and to demonstrate the use of enhanced CPSV-AP through a pilot implementation using a chatbot on top of enhanced CPSV-AP based data that has been organised and stored as a knowledge graph.

Overall, the approach described in the paper is interesting and scientifically valid. The main contribution - in my point of view - is that it attempts to bring together a theoretical piece of work (i.e. the extension of a broadly used vocabulary) with a practical one (i.e. the exploitation of the extended vocabulary/ontology by a knowledge graph and a chatbot using the corresponding data). However, this has not been articulated appropriately and has to be improved. Authors need to better justify their approach and clearly present the contribution(s) of this paper. To do this, the first two sections of the paper need some amendments.

Authors also need to go through the paper and better express statements such as the one appearing in page 1: "We should note here that although a PS description may include a concept termed description, providing a short description of a PS, the term PS description is used to denote a set of concept values".

Section 2.3 (Related work on chatbots and knowledge graphs) has also to be improved to better clarify (i) the structure of a knowledge graph and how a specific ontology can be mapped onto it, and (ii) how chatbots can use (and perform some type of reasoning or QA) the data that are stored in a knowledge graph.

In the last section, authors should clearly discuss the limitations of their work and comment on the application of their approach in other public services. They should also justify statements such as "The use of knowledge graph and chatbot technologies seems very promising in realizing the personalization of PS in real life, large scale scenarios".

Reviewer 3 Report

I appreciate the clarity of title, abstract and Introduction.

Subchapter 2.2.4. Authors mentioned few articles and one book which they identified analyzing similar concepts. The presentation in this subchapter is chaotic, it is difficult to understand which is the background provided by these works and also which is the relevance of these previous findings for the objectives of the article.

Authors stated from the beginning the need to enhance and also the need to extend personalization of the Core Public Service vocabulary. The reader will be curious to understand the validity of this motivations. Which, in my opinion, is not obvious. Being the main reason of writing this article, these elements should be more clear analyzed and presented in the Introduction part.

In subchapter 5.1 is described an example that explains the legitimity of the proposed model. Several other examples of Life Events which require the implementation of the model would support its utility.

The Discussion section lacks in identifying similarities/differences to other models in use/analyzed in other articles.

A Conclusion section is necessary to be included in the article.

Also, the Reference section should be revised.

Round 2

Reviewer 3 Report

I appreciate the changes made by authors. The new version of the article is much improved and, in my opinion, it could be published in the current form